# Female Collegiate Dancers’ Physical Fitness across Their Four-Year Programs: A Prospective Analysis

**DOI:** 10.3390/jfmk8030098

**Published:** 2023-07-17

**Authors:** Jatin P. Ambegaonkar, Jena Hansen-Honeycutt, Kelley R. Wiese, Catherine M. Cavanagh, Shane V. Caswell, Shruti J. Ambegaonkar, Joel Martin

**Affiliations:** 1Sports Medicine Assessment Research & Testing (SMART) Laboratory, George Mason University, Manassas, VA 20110, USA; kwiese2@gmu.edu (K.R.W.); ccavanag@gmu.edu (C.M.C.); scaswell@gmu.edu (S.V.C.); jmarti38@gmu.edu (J.M.); 2Department of Dance, George Mason University, Fairfax, VA 22030, USA; jhansenh@gmu.edu; 3Orthocare Physical Therapy Center, Fairfax, VA 22033, USA; shruti@orthocareptc.com

**Keywords:** performing arts, core, push-ups, planks, single leg hops, training, health, performance, Star Excursion Balance Test, balance

## Abstract

Dance is physically demanding, requiring physical fitness (PF) that includes upper body, lower body, core fitness, and balance for successful performance. Whether PF changes as dancers advance from when they enter (freshmen) to when they graduate from their collegiate program (seniors) is unclear. We prospectively compared collegiate dancers’ freshman-to-senior PF. We recorded PF in regard to upper body strength endurance (push-ups), core strength endurance (front, left-side, right-side, and extensor plank hold times), lower body power (single leg hop—SLH—distances % height; Leg Symmetry Index: LSI = higher/lower × 100, %), and balance (anterior reach balance, % leg length, LL; LSI balance = higher/lower × 100, %) in 23 female collegiate dancers (freshman age = 18.2 ± 0.6 years). Repeated measures ANOVAs (*p* ≤ 0.05) were used to compare measures from freshman to senior years. Across their collegiate programs, dancers’ PF remained unchanged. Specifically, their upper body strength endurance push-up numbers (*p* = 0.93), their core strength endurance plank times (left: *p* = 0.44, right: *p* = 0.67, front: *p* = 0.60, *p* = 0.22), their SLH distances (left: *p* = 0.44, right: *p* = 0.85), and their symmetry (*p* = 0.16) stayed similar. Also, dancers’ right leg (*p* = 0.08) and left leg balance (*p* = 0.06) remained similar, with better balance symmetry (*p* < 0.001) in seniors. Overall, dancers’ PF did not change across their collegiate programs. Thus, female dancers’ freshman PF may be an adequate baseline reference measure when devising rehabilitation programs and determining readiness-to-return-to-activity post injury.

## 1. Introduction

Dancing, like athletics, is a physically challenging activity, but is also unique as it has the added demands of aesthetically appealing artistry [1,2]. Specifically, dancers regularly perform jumping, landing, and other physically demanding movements [1,2,3]. Dancing uses similar components of physical fitness as in athletic sports, [4] placing dancers under a high injury risk [4,5,6]. Accordingly, researchers note that 50–85% of dancers suffer injury during a performance season [5,6,7].

Contrary to the misguided belief that physical attributes detract from the aesthetic of dance, adequate muscular strength, endurance, and power are desirable and necessary qualities for dancers to perform well and prevent injury [2,8]. These parameters, which the American College of Sports Medicine combines into the category of “physical muscular fitness”, can influence dancers’ injury risk [9]. Thus, it is important for dancers to have sufficient physical fitness (PF) comprised of muscular strength endurance, balance, and power to perform well and not get injured when dancing [10,11,12,13].

Dance is taught as a formal collegiate program in over 450 colleges and universities in the United States alone and in many more collegiate dance programs globally [14]. Given that collegiate dancers should be physically fit [15,16] combined with observations of high injury rates in dancers [15,17,18], prior authors have suggested that collegiate dance programs should assess their students’ PF levels [17,19]. Researchers also suggest that practitioners should conduct pre-participation PF screening in dancers [17,20]. This suggestion is in line with athletic settings, where practitioners commonly perform annual pre-participation screenings [21,22]. These screenings have multiple components including general, medical, biochemical, and PF assessments [21]. While researchers note that yearly laboratory and extensive biochemical screenings may not yield substantial clinically significant outcomes from a cost-effectiveness perspective [22,23,24], screenings are still recommended for athletes as they are often among the only formal medical and health assessments for athletes before participation [21].

In a systematic review and meta analysis of screening tools used as injury predictors in dance, Armstrong et al. found some evidence for using dance-specific positions as predictors of dance injury [17]. Kinesiology healthcare practitioners working with dancers also use PF screening measures as baselines when creating rehabilitation and return-to-dance protocols for injured dancers [13]. Kinesiology practitioners conducting assessments could also identify freshman dancers with suboptimal levels of PF. Still, some authors conducting PF screening note that screenings require time and personnel resources [22].

Overall, whether yearly PF testing in collegiate dancers is needed in collegiate dancers is unknown. Relatively little literature [8,16] exists describing PF in collegiate dancers. At present, it is unclear whether female dancers’ PF changes as they progress through their collegiate programs. Finally, whether collegiate program dance training alone is enough to maintain PF also remains unknown. Thus, we prospectively compared collegiate dancers’ freshman and senior PF scores. Our purposes were to describe female collegiate dancers’ PF, examine asymmetries and relationships in PF, and examine whether dancers’ PF changed across their collegiate programs and whether freshman scores are adequate baseline reference measures. Our primary null hypothesis was that dancers’ PF measures would remain unchanged as they progressed from freshman to senior years in a 4-year collegiate dance program. The secondary null hypothesis was that dancers’ PF scores would not be related to each other. Our final aim was to compare dancers’ PF to other studies reporting the same measures in collegiate athletes across other sports.

## 2. Materials and Methods

### 2.1. Study Design

We used a prospective study design to determine PF in undergraduate collegiate dancers over two 4-year periods (2014–2018 and 2015–2019). All PF testing was performed in a single session as part of an annual physical fitness assessment which took place at the start of the academic year for freshmen and at the end of the academic year for seniors by trained investigators. Generally, these were all performed at the same time of day to minimize circadian variation in testing [25]. All investigators followed instructions from the same standard instruction manual and were trained by the same lead investigator in this prospective longitudinal study.

### 2.2. Participants

Twenty-three female collegiate dancers (freshman age = 18.05 ± 0.4 years; dance experience = 13.0 ± 3.3 years; weekly dance training (including classes, rehearsals, and performances) = 26.8 ± 4.2 h) participated in the study. All participants were dance major students in a collegiate program that emphasizes modern dance. However, all dancers had prior experience in other dance styles including, but not limited to, ballet, jazz, and hip-hop dance. The George Mason University Institutional Review Board approved the study (IRB # 736138-5), and all participants gave their written informed consent before participation.

### 2.3. Procedures

A member of the research team led participants through a standard warm-up prior to testing. The warm-up session included dynamic movements to increase heart rate and blood flow to muscles and allow dancers to get ready for testing. The warm-up session included four exercises performed for one minute each. These exercises were as follows: (1) jumping jacks, (2) overhead arm circles in both directions, (3) buttock heel kicks, and (4) mountain climbers. We then recorded the anthropometrics and PF of the lower body, upper body, core, and balance in a station-based format. The primary investigator guided the dancers to the different stations using a randomized format to mitigate fatigue and systematic error effects due to the testing order. All test sessions were conducted by researchers at George Mason University. For all testing sessions, at least two researchers were National-Strength-and-Conditioning-Association-certified strength and conditioning specialists and multiple researchers were state-licensed and National-Board-certified athletic trainers. The PF assessments used in the present study were selected based on the reported test–retest reliability necessary for the annual testing of dancers [26].

### 2.4. Measures

#### 2.4.1. Anthropometrics and Training History

We collected dancers’ anthropometric data: height was measured to the nearest millimeter using a Seca 216 Stadiometer (Scale Co. Inc, Brooklyn, NY, USA) and body mass was measured to the nearest 0.1 kg using a digital scale (Precision Digital Bathroom Scale, HealthTools LLC, Mahwah, NJ, USA) (see Table 1). Dancers also self-reported their dance training history and their types and volumes of weekly supplementary and cross training outside of dance class.

#### 2.4.2. Upper Body Strength Endurance

Dancers’ upper body strength endurance was examined using the push-up test as described in the American College of Sports Medicine’s Guidelines for Exercise Testing and Prescription [9], as follows: The participant assumed a straight leg (for males) or modified “knee push-up” posture in the down position (for females): legs together, lower leg in contact with a mat with ankles plantar-flexed, back straight, hands shoulder width apart, and head up, using the knees as the pivotal point. The dancer then raised the upper body by straightening the elbows until fully extended. Next, the dancer returned to the “down” position by lowering the upper body until the chin touched the mat. The stomach was not allowed to touch the mat, and the dancer’s back was required to remain straight. The test was stopped when the participant strained forcibly or was unable to maintain the appropriate technique within two repetitions. The test has been previously reported to be reliable for examining upper body physical fitness [27].

#### 2.4.3. Core Strength Endurance

We measured dancers’ core strength endurance using plank tests (or bridge tests—used synonymously in the current study) assessing flexor, right, and left lateral sides using procedures described in the prior literature [28,29]. Plank tests have been reported as being valid tests to evaluate core function [30]. For the extensor plank test, we used a modified version of the Biering-Sorensen Extensor Endurance Test [31]. Dancers first performed a single practice trial for a few seconds to confirm that they were able to successfully attain the test position. Then, dancers performed one recorded test trial. For all tests, we recorded the maximum time (s) that the dancers were able to hold and maintain the correct test position.

For the front plank test, dancers assumed a push-up posture in the up position: legs together, toes in contact with a mat with ankles plantar-flexed, back straight, hands shoulder width apart, and head up. We stopped the recording time when any segment of the dancers’ body did not remain parallel to the floor, as described in the prior literature [28].

To perform the left lateral plank test, dancers placed their feet one on top of the other, their left arm (side being tested) perpendicular to the floor, with the elbow resting on the mat and the right arm across the chest on the left shoulder. We used a similar position for the right lateral musculature plank test, with the right arm perpendicular to the floor. The time point when the participants could not maintain a straight line between the trunk or lower body (thigh or shank) segments upon visual observation was recorded by the investigator [32].

For the extensor test, participants lay prone on an examination table with both of their anterior superior iliac spines on the edge of the table, with their hands on the seat of a chair placed in front of them at the edge of the table. A research team member held the participants’ legs above and below their knees to secure the participants’ lower bodies (instead of straps to reduce strap friction discomfort). Time was started when the participants assumed a horizontal position of the trunk, removing their hands off the chair and crossing them across their chest, and stopped when the participants were unable to remain in this position.

#### 2.4.4. Lower Body Power

We examined dancers’ lower body power using the single leg hop (SLH) test as described in the previous literature [33]. Dancers first stood on one leg with their big toe on a marked starting line. Then, they performed a single hop, covering as much distance as possible horizontally, and landed on the same leg. The dancers’ single leg hop distance was measured from the starting line to the heel. Dancers’ arms were unconstrained during the hop. The trial was repeated if (1) the dancer’s contralateral foot touched the floor, (2) the dancer lost balance, or (3) the dancer took additional hops after the single hop. Dancers performed 3 successful trials on each leg, alternating between sides to prevent fatigue. Then, the hop distance was normalized to the dancer’s height (SLH, % height) to reduce the effects of inter-individual anthropometric variations, and we reported these distances for both legs [34]. We averaged the dancers’ SLH across their freshman and senior hop distance measures to calculate average SLH during their collegiate programs.

We also examined hop symmetry across legs. The Leg Symmetry Index (LSI) allows bilateral comparison on a test by calculating a ratio of a low score divided by a high score, which is multiplied by 100 to obtain a percentage [35]. Scores range from 0% to 100%, where 100% suggests full functional symmetry [35]. The LSI is commonly used for return-to-activity decisions with a target of 85% LSI before full return-to-activity [35,36,37], and asymmetries greater than 15% being associated with an increased injury risk [36]. In the current study, the LSI was calculated as (higher value/lower value) × 100 across legs at two time points: freshman and senior scores [38].

#### 2.4.5. Balance

We measured balance using the Star Excursion Balance Test (SEBT) as it is reported to be able to quantify dynamic postural–control deficits caused by lower extremity impairment [39,40]. In the current study, we only used the anterior direction reach of this test, as this direction is reported to be a prospective predictive musculoskeletal injury risk factor in a physically active population [41] and in collegiate athletes [42,43].

We showed participants how to perform the test using both verbal instruction and demonstration, and participants were then allowed 3 practice trials before actual test performance, consistent with previously published instructions [32]. Participants first assumed a single-leg stance, and then maximally reached along marked lines using the other leg while keeping the stance leg stable at the center of a grid, and then returned the reach leg back to the center without losing balance.

Dancers first performed right leg and then left leg reaches. Dancers had a 15 s rest interval between each trial on the same leg, and a 1 min rest interval when changing feet [32]. We did not count a trial and asked the participant to repeat it if (a) the dancer was unable to maintain a single leg stance, (b) the heel of the dancer’s stance foot did not remain in contact with the floor, (c) the dancers’ weight shifted onto the reach foot, or (d) the dancer did not maintain start and return positions each for one second. We averaged and normalized all reach distances across the three trials to % leg length (LL). LL was measured from the anterior superior iliac spine to the medial malleolus. We averaged dancers’ balance across their freshman and senior measures to calculate average anterior reaches during their collegiate programs. We calculated balance asymmetry across legs, as several researchers [41,43,44,45] have used reach asymmetry as an outcome measure of balance. Asymmetry was again calculated using the Leg Symmetry Index, where the LSI = (higher value/lower value) × 100 across legs at two time points: freshman and senior scores [38].

### 2.5. Data Extraction and Statistical Analyses

Data were compiled and inputted into a Microsoft Excel (Microsoft Inc., Redmond, WA, USA) worksheet. The mean and standard deviation of all measures were computed. For the push-ups, the maximum number of push-ups performed consecutively without rest was counted as the score [46]. Dancers’ push-up scores were averaged across their freshman and senior measures to calculate average push-ups during their collegiate programs. The dancers’ core plank times in each direction across their freshman and senior measures were averaged to calculate the average core plank during their collegiate programs. For SLH, we averaged dancers’ SLH across their freshman and senior hop distance measures to calculate average SLH during their collegiate programs.

Repeated measures ANOVAs (*p* ≤ 0.05) compared the outcome measures between the two times (freshmen–seniors). Pearson correlations compared relationships between outcome variables. The relationships’ strengths were operationalized using previous guidelines, where 0.00–0.25 = little or no relationship; 0.26–0.50 = fair relationship; 0.51–0.75 = moderate to good relationship, and 0.76–1.00 = good to excellent relationship [47]. All statistical analyses were performed using Jamovi (Jamovi software, version 0.70) and the statistical significance level was set to *p* < 0.05 for all tests.

## 3. Results

Dancers’ upper body strength endurance, core endurance, and SLH distances and symmetry remained similar from freshman to senior year (see Table 2). Dancers’ right and left side balance remained similar, but their balance symmetry improved from their freshman to senior year over their collegiate program, resulting in better symmetry in senior year balance.

Dancers’ plank times across the multiple directions showed moderate–good positive correlations (r values ranging from 0.6 to 0.9), and the balance reach distances across legs showed good to excellent relationships (r = 0.9). SLH distances across legs were positively correlated (r = 0.8). However, the different PF measures were not related to each other (see Table 3).

## 4. Discussion

In the current study, we examined female collegiate dancers’ PF scores for changes from freshman to senior year. Our findings support the primary null hypothesis as we detected no changes in collegiate dancers’ PF scores. Our secondary hypothesis was partially supported in that the PF measures were positively correlated for tests examining the same body part (e.g., all core PF tests were related) but not correlated over the different PF tests (e.g., core tests were not related to lower leg power).

### 4.1. Upper Body Strength Endurance

The upper body strength endurance (i.e., number of push-ups) of the current dancers values (average ~18–20 push-ups) are similar to previous reports on collegiate dancers [46] and recently published norms in collegiate dancers [13]. The current data suggest that collegiate dancers displayed higher upper body strength endurance as compared to female collegiate soccer players (13.6 ± 2.3) but lower upper body strength endurance than female collegiate gymnasts (30.5 ± 3.4) [48]. These differences could partially be explained considering the differing demands across these activities. Specifically, soccer is a lower-body-intensive sport, while gymnastics is a full body sport that often relies heavily on upper body strength endurance. The dancers in the current program were in a collegiate dance program that emphasizes modern dance. Modern dance has both lower and upper body demands [49]. Additionally, as the dancers are in a formal collegiate program, they also take classes in other dance styles that are lower-body-intensive dance styles like ballet. Therefore, it is understandable that the current dancers’ push-up scores were in between collegiate soccer players and gymnasts.

Interestingly, the average numbers of push-ups of the current dancers were in the “good” range (15–20) of upper body strength endurance, as classified by the American College of Sports Medicine [9]. This finding supports the notion that dancers are aesthetic athletes and need good upper body PF to successfully perform [13].

In the current study, dancers’ upper body strength endurance did not change from freshman year to senior year. This finding suggests that collegiate program dance training helps to maintain but does not enhance upper body PF in collegiate dancers. Depending on the perspective, this observation can be interpreted as either a positive—i.e., dancers stayed fit over their collegiate program—or as a negative, i.e., despite collegiate dancers taking part in a rigorous collegiate program—where they are improving their technical dance skills and expertise as they progress from their freshman to their senior year—their upper body strength endurance does not improve. We propose a possible theory to explain this observation. As dancers become more skilled (progress from being freshmen to seniors), they become more efficient and can perform higher-skill movements without needing to use their maximal PF capacity to maintain aesthetics. In support of this theory, dancers have been previously reported to subconsciously downmodulate jump heights and maximal electrical muscle activity of the quadricep muscles likely in an effort to maintain the aesthetic component of dance [50]. We recommend additional research to verify this theory to examine whether as dancers progress in their careers (from freshmen to seniors and beyond), they subconsciously downmodulate their maximal PF values in the upper body, lower body, in functional activities, and in dance-specific movements.

Prior researchers have found a high incidence of shoulder injuries in collegiate modern dancers and suggested that upper body fitness should be included in pre-participation physical screening [49]. However, the authors did not examine baseline upper body strength measures. The current findings support the need to create norms for upper body strength endurance in dancers across different genres (e.g., hip-hop, Indian classical dance, jazz) and levels (e.g., adolescents, professional dancers). This work can help rehabilitation clinicians to determine initial pre-injury PF levels and they can use these values as appropriate target rehabilitation cut-offs when working with injured dancers [13].

### 4.2. Core Strength Endurance

We chose plank tests to examine core PF in the current study as they are commonly used in the literature, are reliable, are valid global core muscle fitness measures, and are easy to administer with low technical demands [29,51]. The current dancers’ average core flexor plank scores (~116 s) are similar to prior reports on collegiate students (~108 s) [52] but somewhat lower than past ranges (~149–171 s) in healthy collegiate females [29], collegiate dancers [53], and resistance-trained females [54]. The dancers’ right and left plank core strength endurance scores (~66–72 s) are similar to previous observations in collegiate dancers [53], college students [52], healthy collegiate females [29], and resistance-trained females [54], but somewhat lower than those of elite state-level athletes (~80–85 s) [51]. The current dancers’ extensor plank scores (~131–136 s) are somewhat lower than reports on female cross-county athletes (151 s) [55] and elite state-level athletes (~164 s) [51], and are similar to college students (125 s) [52].

Consistent with prior observations [53], the current dancers’ core strength endurance plank scores had large standard deviations, possibly due to the nature of the plank tests which allowed participants to use different strategies to maintain test positions. Prior authors [53] have noted that multiple muscles may influence plank test results: for example, different dancers may use their shoulder and leg musculature differently to maintain their bodies in the plank position, suggesting the need to consider other tests to examine core function. Furthermore, core stability exists in a continuum where the core muscles need to produce increasing amounts of force over decreasing amounts of time from core endurance to strength to power [56]. Also, other tests exist to examine core PF, including the bent knee lowering test [57], among others. Thus, researchers examining core PF should consider both types of tests: ones that isolate the core muscles, and ones that include core and other muscle activity to perform whole-body functional movements.

Overall, the current results support reports that dancers require good core PF to allow them to successfully perform while not getting injured [56]. Therefore, researchers should establish baseline core strength endurance norms in collegiate and other dancers to aid kinesiology practitioners in choosing appropriate norms to grade their dancers’ baseline PF based on their participants’ levels and task demands.

### 4.3. Lower Body Power

The present findings indicate no changes in SLH distances from freshman to senior year in collegiate dancers. In dancers, one group of authors [11] found that dancers whose SLH distances were less than 78.2% of their height had an increased risk of lower body injury. However, in athletes, the majority of the research on SLH tests focuses on recovery from anterior cruciate ligament reconstruction surgery [33,58,59]. To our knowledge, we could not find research examining changes in SLH in the same individuals over time. Thus, future research is needed to investigate this gap in the literature and clarify whether regular participation in physical activity influences PF changes in single leg hop distances over time in dancers and athletes.

The current dancers’ SLH scores (~81% BH) are similar to those reported in collegiate dancers (80–81% BH) [11,34] but lower than some reports on 18–29 year old adults (~89% BH) [60] and 15–16 year old soccer athletes (99% BH) [61]. The differences in scores could partially be explained by differences in training and task demands. Clinically, these findings indicate that dancers’ lower body strength should not be directly compared to lower body power norms in sports and support the need to establish dancer-specific norms for lower body power.

We also recognize that PF can be examined using different outcome measures; for example, power can be measured using a variety of hop tests, including the triple leg hop for distance and multiple hop tests. Still, some recent evidence exists noting that in dancers, SLH performance is positively correlated with balance [34] and can prospectively predict lower body injury risk [11]. Dancers also regularly perform hops and jumps within class, rehearsal, and performance [62]. Overall, we recommend the use of the SLH test as an easy-to-administer-and-interpret test in dancers as a measure of lower body power and PF.

### 4.4. Balance

The current dancers’ left and right anterior reach balance scores (~75% LL) are similar to ranges in collegiate dancers [34] and active adults [63] (~75% LL) but lower than for female lacrosse players (96% LL) [57]. A possible reason for this difference could be different exercise regimes and different demands of the sports. We chose the anterior reach distance to examine dancers’ balance as it has been reported to predict injury risk [42,43]. Thus, we are reasonably confident in this test as a measure of balance.

Although the anterior reach distance may be an appropriate balance measure, some authors have cautioned against only using this anterior distance reach or only using the Y-balance test as predictive injury measures [45]. This is because the Star Excursion Balance Test or its modification—the Y-balance test—may not conclusively predict injury risk in all athletes [38]. Variations of the Star Excursion Balance Test exist in dancers [64,65,66] but these do not appear to have robust face or content validity as balance screenings [65] or are not different from the Star Excursion Balance Test when controlling for upper body motion [66]. Thus, the anterior reach direction may offer a better option for kinesiology practitioners examining balance in their dancers, as it can be conducted in less time and has low technical resources and effort needs.

When considering the LSI, previous researchers have suggested that 3–8% asymmetry (i.e., 93–98% LSI) in dynamic balance tests is normal in healthy adults [35]. The current dancers’ LSIs were within these ranges, both when they were freshmen and when seniors, indicating that the dancers were functionally symmetrical in their lower body power PF. Still, the dancers’ balance symmetry improved from their freshman to senior year (92% to 97%), that is, their balance was more symmetrical when they were seniors. This observation is encouraging as it indicates that there was no lateral biasing in the current dancers during their time in the collegiate program. This lateral biasing due to dance training has previously been suggested to be related to dance injuries [67].

Another point to note regarding asymmetry is the caution suggested by some authors [68] in not using a single cut-off point (e.g., difference of >4 cm across sides) to determine injury risk. This is because injury risk is often multi-factorial, population-dependent, and not generalizable across different populations. Thus, future researchers should continue to examine leg asymmetries and individual balance reach distances when using the Star Excursion Balance Test, the Y-Balance test, or its components as part of a larger comprehensive assessment of PF in dancers.

### 4.5. Relationships among Physical Fitness Measures

The current dancers’ different PF measures were understandably positively correlated across sides. Specifically, the four core plank tests were positively correlated to each other, and the balance reach distances and hop distances across legs were positively correlated. However, the different PF measures were not related to each other. This finding implies that PF differs across different body regions and may be demand- and task-dependent. Thus, practitioners should use multiple tests that test individual body parts like plank tests for the core, and also functional tests like hop tests which examine overall body function during movement as components of a larger comprehensive assessment of PF in dancers across different genres. Using this comprehensive approach can provide kinesiology practitioners with a holistic perspective regarding dancers’ PF in their natural settings, not limited to laboratory settings.

### 4.6. Physical Fitness and Supplementary Training in Collegiate Dance Programs

The current finding that dancers’ PF remains unchanged over a 4-year period in collegiate dance programs indicates that regular collegiate dance training practice may assist in the maintenance of PF in dancers. However, dance training alone may not improve PF. Rather, added supplementary training may be needed to enhance dancers’ performance as they progress in their program [69]. Collegiate dance programs are suggested to include some consistent form of PF and supplementary training for their dancers [70]. In a study examining health literacy in collegiate dancers and collegiate dance programs, Kozai and Ambegaonkar [70] found that while several collegiate programs discussed some PF aspects in their curricula, not all dance programs provided formal curricular instruction about PF training in the program [70].

Also, while some collegiate programs had dedicated healthcare support [71], how much training and conditioning is offered in these programs is still unclear. In a systematic review examining the effects of supplementary training in dance, the researchers [69] found that, generally, in female collegiate dancers, supplementary training does enhance dancers’ performance, with limited evidence for injury risk reduction. Specifically, the researchers found that supplementary PF training programs that included multiple exercises, lasted around 1 h per session, and were implemented 2 to 3 times per week for around 8 weeks enhanced dancers’ performance, and had limited evidence of reducing dancers’ lower back pain and injury risk.

The current dancers indicated that they took part in an average of ~2 h/week of supplementary PF training outside of their formal curriculum dance classes. The training included combinations of conditioning, Pilates, jogging, CrossFit training, and other PF training regimens. However, this training was not part of the formal dance program. Also, this training was not designed or supervised by a trained practitioner. Thus, we are unsure about the rigor and intensity of these training sessions and cannot state as to whether these training programs had any specific effect on the dancers’ PF. Based on the current findings, we can speculate that dance alone may not elicit a strong enough stimulus to induce significant PF changes without additional supplementary fitness training. As discussed in the sections above, the PF levels of the dancers were comparable to athletes in other sports. Thus, their baseline fitness levels were indicative of relatively high PF levels that would could be further assisted with targeted fitness programs that incorporate basic design principles of specificity and progressive overload to promote further positive physiological adaptations [72]. Accordingly, future researchers should examine whether supplementary PF training interventions designed and implemented by trained practitioners can enhance the PF measures used in the current study to further optimize dancers’ PF and performance.

### 4.7. Limitations and Future Recommendations

We recognize that our findings have limited generalizability outside of our study participants. Nevertheless, the current PF scores are in general agreement with prior PF measures. We also recognize the low sample size and small effect sizes. Still, to our knowledge, the study is novel as it is among the first ones to prospectively examine these PF measures in female collegiate dancers.

It is also important to note that the several PF measures used in the current study are relative measures that depend on the individuals’ anthropometrics and are not absolute measures. These normalizations of individuals’ measures provide a more representative measure and reduce the effects of individual anthropometric variations [34]. Still, practitioners should be aware about whether the measures that they examine are relative or absolute measures when comparing them to other published reports. Still, we suggest that others can use the tests in the present study as they are low in technology needs, can be performed in dance studios, and do not need extensive technical training.

The current results indicate that the dancers’ PF did not change from when they were freshmen to when they were seniors. However, there could have been changes in the dancers’ PF each year (freshman–sophomore–junior–senior). However, we only collected data at two time points: when the dancers entered the program (freshmen) and when they graduated from the program (seniors). Thus, we cannot speculate regarding how the dancers’ PF changed across time for each year in the program and we can only discuss PF over the two time points (freshman and senior) examined in the current study. We suggest that practitioners consider yearly testing if they are able to successfully overcome the personnel and timing logistical challenges, so that they can examine year-to-year changes in female collegiate dancers’ PF over time.

We also recognize that there are logistical challenges including time and personnel resources when collecting injury and exposure records; in agreement with prior authors’ suggestions, we recommend that researchers may explore the relationship between PF exposure and injury epidemiology in female collegiate dancers. Specifically, future researchers should examine whether any of these PF measures taken at the start of the freshman year (or beginning of the training program year) can assist practitioners in prospectively predicting injury incidence in dancers across different levels and genres.

Also, using normative values is not the only measure that practitioners can use when considering injury prevention or rehabilitation post injury. Rather, we suggest developing normative values as baselines for healthy dancers, like norms in other physically active populations, so that practitioners can use these values to recognize possible high-risk dancers, implement training programs, and determine safe return-to-activity benchmarks. Finally, multiple variables can influence changes in fitness across time. Thus, we recommend larger prospective and multi-institutional examinations so that researchers can divide dancers into further subgroups—e.g., in whom fitness remained unchanged, in whom fitness improved, and in whom fitness reduced—and examine the effects of individual variables such as training, age, injury, and environment on dancers’ PF over time.

### 4.8. Practical Implications

The practical implications of the current study are to inform multiple stakeholders in the kinesiology field. Specifically, this evidence can support dancers and dance educators in proactively including supplementary training and conditioning programs as an inherent part of dance training in female collegiate dancers. The current results that dancers’ freshman-year PF remains generally similar through to senior year agrees with prior suggestions in professional athletes that yearly pre-season laboratory screening would not be warranted under normal clinical standards [22]. Strength and conditioning practitioners working with collegiate dancers can use freshman PF screenings as baselines and will not need to repeat this testing when setting training goals. Kinesiology practitioners can use this information to determine return-to-activity in their dancers when devising rehabilitation programs for their injured collegiate dancers. Finally, the findings that female dancers’ PF scores were generally lower than those reported in athletes but were similar to recreational and healthy populations indicate support for different norms for different groups of physically active populations [13].

## 5. Conclusions

In conclusion, the findings of this study suggest that female dancers’ PF remains consistent throughout their college program, with the exception of improved balance scores, which could have important implications for injury prevention and performance enhancement in this population. This finding suggests that female dancers’ freshman scores may be adequate baseline reference measures when devising rehabilitation programs and determining readiness for return-to-activity post injury.

## Figures and Tables

**Table 1 jfmk-08-00098-t001:** Anthropometrics of female collegiate dancers (means ± standard deviation).

Measure	Side	Freshmen	Seniors	F-Value	*p*-Value	Effect Size
Height (cm)		161.6 ± 4.9	161.8 ± 4.9	0.9	0.32	0.21
Weight (kg)		57.3 ± 6.0	58.0 ± 5.7	0.83	0.37	0.20

**Table 2 jfmk-08-00098-t002:** Physical fitness of female collegiate dancers (means ± standard deviation).

Measure	Side	Freshmen	Seniors	F-Value	*p*-Value	Effect Size
Height (cm)		161.6 ± 4.9	161.8 ± 4.9	0.9	0.32	0.21
Weight (kg)		57.3 ± 6.0	58.0 ± 5.7	0.83	0.37	.20
Push-Ups (number)		18.5 ± 4.7	18.9 ± 6.1	0.09	0.76	0.07
Core Plank Test Times (seconds)						
	Flexor	116.2 ± 46.0	112.2 ± 47.1	0.19	0.66	0.10
	Left	69.0 ± 35.4	72.9 ± 19.2	0.27	0.60	0.11
	Right	66.7 ± 32.4	70.7 ± 15.1	0.48	0.50	0.15
	Extensor	136.8 ± 29.2	131.4 ± 39.7	0.29	0.59	0.12
Single Leg Hop Distance (×body height)						
	Left	80.6 ± 11.4	86.1 ± 10.0	2.84	0.11	0.38
	Right	81.6 ± 8.3	86.1 ± 8.4	1.88	0.19	0.31
LSI Single Leg Hops (%)		95.6 ± 3.7	96.5 ± 1.9	1.19	0.29	0.25
Anterior Reach Balance Distance; % Leg Length						
	Right	71.8 ± 6.5	74.7 ± 6.3	2.06	0.17	0.32
	Left	71.5 ± 6.6	75.7 ± 5.9	3.87	0.06	0.44
LSI Balance (%)		91.6 ± 5.3	97.0 ± 2.6	15.3	8.11 × 10^−4^	0.83

**Table 3 jfmk-08-00098-t003:** Correlation analysis matrix of physical fitness in female collegiate dancers.

		Flexor Plank	Right Plank	Left Plank	Extensor Plank	Balance— Right	Balance— Left	SLH— Left	SLH— Right	Push-Ups
Flexor Plank	r	—								
	*p*	—								
Right Plank	r	0.6 **	—							
	*p*	0.0031	—							
Left Plank	r	0.39	0.9 ***	—						
	*p*	0.08	6.70 × 10^−7^	—						
Extensor Plank	r	0.3	0.4	0.4	—					
	*p*	0.2	0.05	0.08	—					
Balance—Right	r	0.06	0.4	0.2	0.2	—				
	*p*	0.81	0.05	0.3	0.5	—				
Balance—Left	r	0.11	0.4	0.24	0.2	0.9 ***	—			
	*p*	0.6	0.09	0.3	0.4	6.04 × 10^−9^	—			
SLH—Left	r	−0.1	−0.1	−0.2	0.04	0.3	0.4	—		
	*p*	0.6	0.8	0.5	0.9	0.26	0.13	—		
SLH—Right	r	−0.3	−0.2	−0.3	−0.2	0.17	0.3	0.9 ***	—	
	*p*	0.16	0.4	0.2	0.5	0.5	0.3	3.28 × 10^−8^	—	
Push-Ups (n)	r	0.4	0.1	0.18	−0.02	−0.1	−0.2	−0.13	−0.2	—
	*p*	0.1	0.5	0.4	0.9	0.8	0.5	0.6	0.4	*—*

**Note**. ** *p* < 0.01, *** *p* < 0.001; r = Pearson’s r value, *p* = -value; flexor plank = flexor plank time (s), right plank = right plank time (s), left plank = left plank time (s), extensor plank = extensor plank time (s), balance—right = anterior reach balance distance—right (% leg length—LL), balance—left = anterior reach balance distance—left (% LL), SLH—left = single leg hop distance—left (body height, BH), SLH—right = single leg hop distance—right (body height).

## Data Availability

The data and code that support the findings of this study are available from the corresponding author, J.P.A., upon reasonable request.

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
