# Peer review of "Female Collegiate Dancers’ Physical Fitness across Their Four-Year Programs: A Prospective Analysis"

_jfmk, 2023, doi:10.3390/jfmk8030098_

Round 1

Reviewer 1 Report

Please see attached

Minor language issues were detected. 

Author Response

Abstract

Line 16: n=number is unnecessary. You can include only the number of push-ups in the parenthesis.

Response: Removed

Line 17: s=seconds is also unnecessary as we understand that hold time is a timed measurement.

Response: Removed

Lines 20-21 need to be presented somewhat differently. “upper body strength-endurance push-ups numbers(p=.93), core strength-endurance plank times”. I understand what you mean, but the flow and structure of the whole sentence need to be improved.

Response: Revised

The recommendation at the end of the abstract is ok, even though you hadn’t stated before that the reason for this assessment was to provide aid or guidelines for rehabilitation purposes. I am not sure that the following statement can be supported based on the purpose of the study (Findings suggest that kinesiology practitioners can use collegiate dancers’ freshmen baseline PF when devising rehabilitation programs and making return-to-activity decisions post-injury throughout their dance programs). Was that the purpose of this project?

Response: Thank you and we agree. We have revised based on purpose as per reviewer suggestions.

Methods

Considering the significant differences in the performance of male and female participants, I am not sure you can justify using only two male subjects in the study.

Response: Revised

The procedures and tests are well-presented, described and supported by previous research.

Response: Thank you

Results

Results are correctly presented, but considering the inclusion of two male subjects, I am unsure how the means were affected.

Response: Revised

Discussion

Lines 245-246 You stated: The upper body strength-endurance (i.e., number of push-ups) of the current dancers’ values (average ~ 19-20 push-ups) are similar to previous reports in collegiate dancers [50] and recently published norms in collegiate dancers [14]. Males, females or both?

Response: Revised to refer to only females

Lines 247-248, you stated: The current data suggest that collegiate dancers displayed higher upper body strength-endurance as compared to female collegiate soccer players (13.6 ± 2.3). This statement is not accurate, considering you included two male subjects.

Response: Revised

The discussion has several inconsistencies due to including 23 female and two male participants. I am not sure why you included the two male subjects, as it doesn’t change your power much; it just complicates your analysis and discussion

Response: Revised

Reviewer 2 Report

In their manuscript, Ambegaonkar et al. set out to present anthropometric measures and physical fitness (PF) of collegiate dancers during their 4-year collegiate career. The authors do a fine job of presenting clear and concise methods based on the ACSM testing guidelines. I support the acceptance of this manuscript, so long as my 5 major concerns below can be addressed. By addressing these concerns, the authors will provide greater transparency in their results/statistics. These changes will positively impact the readers experience, while ensuring that the authors’ conclusions accurately reflect their observations.

1.      The primary aim of the research was to describe collegiate dancer physical fitness (PF) throughout their 4-year collegiate career. However, the results only provide Freshman, Senior, and Average data. The average data is of little use, due to time as a confounding variable. The authors must provide the statistics for anthropometric and PF measures across time (Freshman – Sophomore – Junior – Senior). A repeated measures analysis of variance is the appropriate test to compare values over time.

2.      While most of the anthropometric and PF remained unchanged over time (P > 0.05), the authors must discuss the power of each statistical test to address the possibility of a type II statistical error.

3.      If the authors have any access to injury records, they may explore the relationship between PF and injury frequency. This is a large part of their rationale for measuring PF but they provide no reference, or injury background of their dancers. This is important for the readers to have a better understanding of the study sample, as it compares to the general study population and directly ties back to the authors justification/rationale for the research project.

4.      There is a very large discrepancy in the number of female (23) vs male (2) dancers. It would be appropriate to leave the males out of this manuscript as their number is insufficient to provide any meaningful sex-based differences, while potentially increasing the variability of the measures.

5.      It must be noted that ALL of the strength and endurance measures are measures of RELATIVE strength and endurance, which depend on body size (there a possible trend of increasing body size when comparing Freshman – Senior). There are no absolute strength/endurance measures. Please discuss this and the implications in the discussion.

Minor:

Line 43: The words influence and affect are both used, in succession. This error can be corrected by choosing one over the other, based on author preferences.

Quality of English is acceptable. Only minor spelling errors that can be fixed with a quick proof-read

Author Response

In their manuscript, Ambegaonkar et al. set out to present anthropometric measures and physical fitness (PF) of collegiate dancers during their 4-year collegiate career. The authors do a fine job of presenting clear and concise methods based on the ACSM testing guidelines. I support the acceptance of this manuscript, so long as my 5 major concerns below can be addressed. By addressing these concerns, the authors will provide greater transparency in their results/statistics. These changes will positively impact the readers experience, while ensuring that the authors’ conclusions accurately reflect their observations.

  1. The primary aim of the research was to describe collegiate dancer physical fitness (PF) throughout their 4-year collegiate career. However, the results only provide Freshman, Senior, and Average data. The average data is of little use, due to time as a confounding variable. The authors must provide the statistics for anthropometric and PF measures across time (Freshman – Sophomore – Junior – Senior). A repeated measures analysis of variance is the appropriate test to compare values over time.

Response: Revised – conducted repeated measures analysis of variance and removed the average data.

  1. While most of the anthropometric and PF remained unchanged over time (P > 0.05), the authors must discuss the power of each statistical test to address the possibility of a type II statistical error.

Response: Revised - added effect sizes and discussed in the limitations section.

  1. If the authors have any access to injury records, they may explore the relationship between PF and injury frequency. This is a large part of their rationale for measuring PF but they provide no reference, or injury background of their dancers. This is important for the readers to have a better understanding of the study sample, as it compares to the general study population and directly ties back to the authors justification/rationale for the research project.

Response: We agree. We have added this as a point in the revised discussion and plan to include injury records in future projects. Specifically, we recommend that researchers may explore the relationship between PF and injury frequency in female collegiate dancers.

  1. There is a very large discrepancy in the number of female (23) vs male (2) dancers. It would be appropriate to leave the males out of this manuscript as their number is insufficient to provide any meaningful sex-based differences, while potentially increasing the variability of the measures.

Response: Removed males

  1. It must be noted that ALL of the strength and endurance measures are measures of RELATIVE strength and endurance, which depend on body size (there a possible trend of increasing body size when comparing Freshman – Senior). There are no absolute strength/endurance measures. Please discuss this and the implications in the discussion.

Response: Added this information in the revised discussion.

Minor: Line 43: The words influence and affect are both used, in succession. This error can be corrected by choosing one over the other, based on author preferences.

Response: Thank you. Removed the word affect.

Reviewer 3 Report

Dear Author

Thank you for submitting to JFMK. And thank you for the opportunity to review.

Unfortunately, this study has many shortcomings.

First of all, the sample size of 25 people for on-topic analysis is too small.

In particular, the study design is not fresh at all and the analysis is not appropriate.

There are many things that influence changes in fitness, such as training, age, injury, and environment. It is natural that only 4 years of time does not change the fitness of athletes or dancers who train continuously.

Moreover, the correlations among the fitness variables analyzed in this study are very low in relation to the subject of this study (Table 2).

I recommend the following research.

The authors divide the group into a group in which fitness is maintained or improved for 4 years and a group in which fitness is reduced. And, it is suggested that it is more appropriate to look for the causes of decline in fitness and compare differences in their injury rates and performance.

It needs to be improved more as an academic expression than a grammatical problem.

Author Response

Thank you for submitting to JFMK. And thank you for the opportunity to review.

Unfortunately, this study has many shortcomings.

First of all, the sample size of 25 people for on-topic analysis is too small.

Response: Thank you for the review. In the revised manuscript, we note the low sample size.

In particular, the study design is not fresh at all and the analysis is not appropriate.

Response: The study design is a prospective design. We have revised analyses based on other reviewer suggestions and use a repeated measures analysis of variance. We have added that to our knowledge the study is novel as it is among the first ones to prospectively examine these PF measures in female collegiate dancers.

There are many things that influence changes in fitness, such as training, age, injury, and environment. It is natural that only 4 years of time does not change the fitness of athletes or dancers who train continuously.

Response: We agree. Yet, it may always not be ‘natural’ that dancers’ PF does not change, as for some dancers fitness may improve, for others it may remain consistent, and for some it may worsen as the progress in their program – which was what we were examining in the study. Per reviewer suggestions, we have added information about using prospective and larger study designs to examine effects of individual variables on dancers’ fitness over time.

Moreover, the correlations among the fitness variables analyzed in this study are very low in relation to the subject of this study (Table 2).

Response: Thank you. In revision, we have added that the relationships’ strength was operationalized using previous guidelines, where 0.00-0.25 = little or no relationship; 0.26- 0.50 = fair relationship; 0.51-0.75 = moderate to good relationship, and 0.76-1.00 = good to excellent relationship (Portney 2009).

I recommend the following research.

The authors divide the group into a group in which fitness is maintained or improved for 4 years and a group in which fitness is reduced. And, it is suggested that it is more appropriate to look for the causes of decline in fitness and compare differences in their injury rates and performance.

Response: We understand and agree with the concept for future research. Also, we note the reviewer’s comment on the small sample size. Thus, dividing into further subgroups (e.g., fitness is maintained, fitness is improved, and fitness is reduced) will further reduce sample sizes and another factor in the statistical analyses. Still, we thank the reviewer for the good suggestion. We have added this in future recommendations in the revision as such: we recommend larger prospective and multi-institutional examinations so that researchers can divide dancers into further subgroups in whom fitness remained unchanged, in whom fitness improved, and in whom fitness reduced - and examine the effects of individual variables such as training, age, injury, and environment on dancers’ PF over time.

We do not have injury rates and so cannot report in the current study and have added this information in the limitations and recommendations section.

Overall, we thank the reviewer for their valuable input.

Round 2

Reviewer 1 Report

I am happy with the changes and have no further concerns or recommendations. 

Author Response

Thank you for your constructive comments

Reviewer 2 Report

The authors failed to address one of my major concerns and did not include the data across time for each testing year (freshmen - sophmore - junior - senior). They only removed the group averages (confounded by time) but did not show the data across time for all testing years. Major lack of transparency

Author Response

We did not collect data every year. We only collected data when the dancers entered the program (freshman) and when they graduated from the program (senior). We have added clarity in the methods section of the revision.

We understand the reviewer’s point and have added the following information in the limitations section of the revision as follows:

The current results indicate that dancers’ PF does not change from when they were freshmen to when they were seniors. However, there could have been changes in the dancers’ PF each year (freshmen - sophomore - junior - senior). However, we only collected data at two time points: when the dancers entered the program (freshman) and when they graduated from the program (senior). Thus, we cannot speculate how the dancers' PF changed across time for each year in the program and can only discuss PF over the two-time points (freshmen and seniors) examined in the current study. We suggest that practitioners consider yearly testing if they are able to successfully overcome the personnel and timing logistical challenges so that they can examine year-to-year changes in female collegiate dancers’ PF over time.

Reviewer 3 Report

Thank you for your efforts to improve.

If possible, I hope this document has an analysis to insert one more table to increase its academic value. This is a recommendation.

I have no comments. 

Please take a look at it once more for the sake of academic expression.

Author Response

Thank you Per recommendation, we have added 1 more table for dancers’ anthropometrics

Round 3

Reviewer 2 Report

Thank you for clarifying the pseudo-longitudinal design of the study. It is more closely aligned with a pre- and post- study design, which is now more clear.